# Japan's Productivity and GDP Growth: The Role of Private, Public and Foreign R&D 1967–2017

**THW Ziesemer** 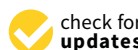

Department of Economics and UNU-MERIT, Maastricht University, 6211AX Maastricht, The Netherlands;
T.Ziesemer@maastrichtuniversity.nl

**Abstract:** We analyze the dynamic interaction of Japan's total factor productivity, gross domestic product (GDP) domestic and foreign private and public research and development (R&D) in vector-error-correction models (VECMs) for Japan with data from 1963–2017. Extensive testing leads to favoring a model with five cointegrating equations for the six variables. Analysis of effects of permanent policy changes shows that (i) additional public R&D encourages private R&D and total factor productivity (TFP), and has higher internal rates of return than private R&D changes and therefore could speed up Japan's growth; (ii) public R&D changes have a statistically significant positive permanent effect on foreign private R&D stocks and a transitional effect on foreign public R&D stocks; (iii) private R&D changes have a statistically significant positive permanent effect on foreign public R&D stocks and a transitional effect on foreign private R&D stocks; (iv) after a temporary GDP change, public R&D is counter-cyclical in the short and medium run and private R&D is pro-cyclical. Empirical results are related to the parameters of a VES (variable elasticity of substitution) function for TFP production.

**Keywords:** public and private R&D; productivity; growth; spillovers; vector-auto-regression/error-correction (VAR/VECM)

## 1. Introduction

Since 1992, Japan's growth rate of GDP or GNI (gross national income) per capita has been lower than those of other high-income countries, OECD (Organization for Economic Cooperation and Development) members, members of the EURO area, or the USA[1]. As this is a long period of almost thirty years, policies should address total factor productivity (TFP) and the factors driving it, R&D and human capital. In this paper, we focus on R&D and discuss closely related literature on Japan's human capital. Dynamic analyzes of the effect of R&D on TFP have considered domestic private and public (non-business) and foreign total R&D stocks (Eaton and Kortum 1997; Luintel and Khan 2004). Most recently, Soete et al. (2020a, 2020b) distinguished between public and private foreign R&D stocks in a policy analysis for the Netherlands and for 17 countries. In this paper, we also use these four R&D variables to analyze the TFP–R&D links and growth of Japan. The domestic public R&D stock[2] is an important policy variable for the role of governments in stimulating the growth process. The research questions are the following. Does public/non-business R&D affect business R&D? Do the

---

[1] For earlier periods, see Fukao et al. (2019).
[2] Public R&D stocks are obtained from accumulation of GERD-BERD (gross expenditure on R&D minus business expenditure on R&D) as explained below.

private and public R&D stocks affect TFP and GDP? Do the returns, which are positive if the above effects are positive, outweigh the costs? Does Japanese R&D affect foreign R&D? Does foreign R&D affect Japanese R&D? As these are empirical questions, we use a dynamic method that lets the data speak, the vector autoregressive or cointegrated VAR approach, rather than imposing the effects by assumption as in theoretical models. Lau (2008) has shown that exogenous and endogenous growth models may be special cases of a VECM. Estimating a VECM and then finding what the effects are gives an answer to the question whether the model is one of exogenous or (semi-) endogenous growth.

We consider these issues for Japan and explain the country-specific aspects in due course below. In Section 2 we report the data and some properties which merit special attention. In Section 3 we introduce the vector–error–correction model, related econometric concepts, and the concept of internal rate of return. In Section 4 we consider results regarding unit roots, cointegration, three VECMs, their comparison in the light of critical testing including absence of (weak) endogeneity, and their link to the literature regarding growth of Japan. More literature is discussed in Section 4.3 in order to show that there is nothing similar to the plan of this paper. To understand this, it is necessary that we explain our approach in Sections 2–4 and then compare it to the literature. In Section 5.1 we analyze effects of changes in R&D policies, public and private, on TFP, GDP and foreign R&D, showing many years of positive effects, the positive average gains/GDP ratio, and their high discounted net present values, and very high internal rates of return based on quickly obtained returns. In Section 5.2, we analyze the effects of changes in policies of foreign public and private R&D and Japan's reaction according to our model. Section 5.3 shows that temporary changes of GDP have countercyclical effects for domestic public R&D in Japan, but pro-cyclical effects for private R&D. Section 5.4 links two of the cointegrating equations to the parameters of a generalized CES or VES function for TFP from a recent growth model. Section 6 summarizes the lessons and concludes the paper.

## 2. Data

Our data are arranged from the perspective of public and private performance rather than financing. We take the total flows of business and non-business R&D in $2005, PPP. From these we construct stock values using the perpetual inventory method with a standard rate of depreciation of 15% (Hall et al. 2010; Luintel et al. 2014). Non-business R&D (henceforth called 'public', abbreviated as PUBST) is calculated as total R&D (GERD) minus business R&D (BERD, henceforth called 'private'). For each country, there is a distance-weighted average of public and private foreign R&D stocks. Domestic and foreign private and public R&D stocks (abbreviated as BERDST, FBERDST, PUBST, FPUBST) are taken from the UNU-MERIT database (assembled from all vintages of publicly available OECD MSTI data including those in book publications) as in Soete et al. (2020a, 2020b) but extended to cover three additional periods[3]. These data are available for the period 1963–2017. GDP data are taken from World Development Indicators and are transformed into 2005 PPP dollars. TFP data are from PWT9; they also go until 2017 now and are strongly revised for Japan compared to those ending in 2014. They do not include human capital (Feenstra et al. 2015). We do not use human capital as a separate variable because it is included in R&D expenditure flows[4]. There have been revisions in Japan's R&D data in 1996, 2008, 2013. We use unit dummies for the periods before the revisions. Johansen et al. (2000) argue that the standard methodology of Johansen (1996, chapter 10) applies in the presence of these dummies. Figure 1 shows that all growth rates were strongly falling over time until about 1995 followed by ups and downs. However, the growth rates of GDP and business R&D

---

3　Special thanks go to Bart Verspagen for providing these data from the UNU-MERIT database.
4　In contrast, OECD measures of TFP do include human capital as only labour and capital are subtracted from GDP. As R&D also includes human capital, a regression of both including a third human capital variable as found in some articles is likely to be strong because of the common human capital data included and may lead to collinearity.

remained positive in the nearest neighbor fit line[5], whereas those of public R&D are zero from 2000 to 2020, and that for TFP is temporarily zero for many years, all with much variation. Japan's ratio of public to private R&D stocks has fallen slightly from 0.97 in 1963–1981 to 0.91 in 2017. Archibugi and Filippetti (2018) have criticized similar trends away from public to relatively more private financing and performance of R&D in the corresponding flow data for G7 countries. For the case of Japan the question is whether growth rates should be more in proportion for public and private R&D, requiring a policy revision. Growing private R&D alone has not prevented the fall of TFP growth to roughly zero.

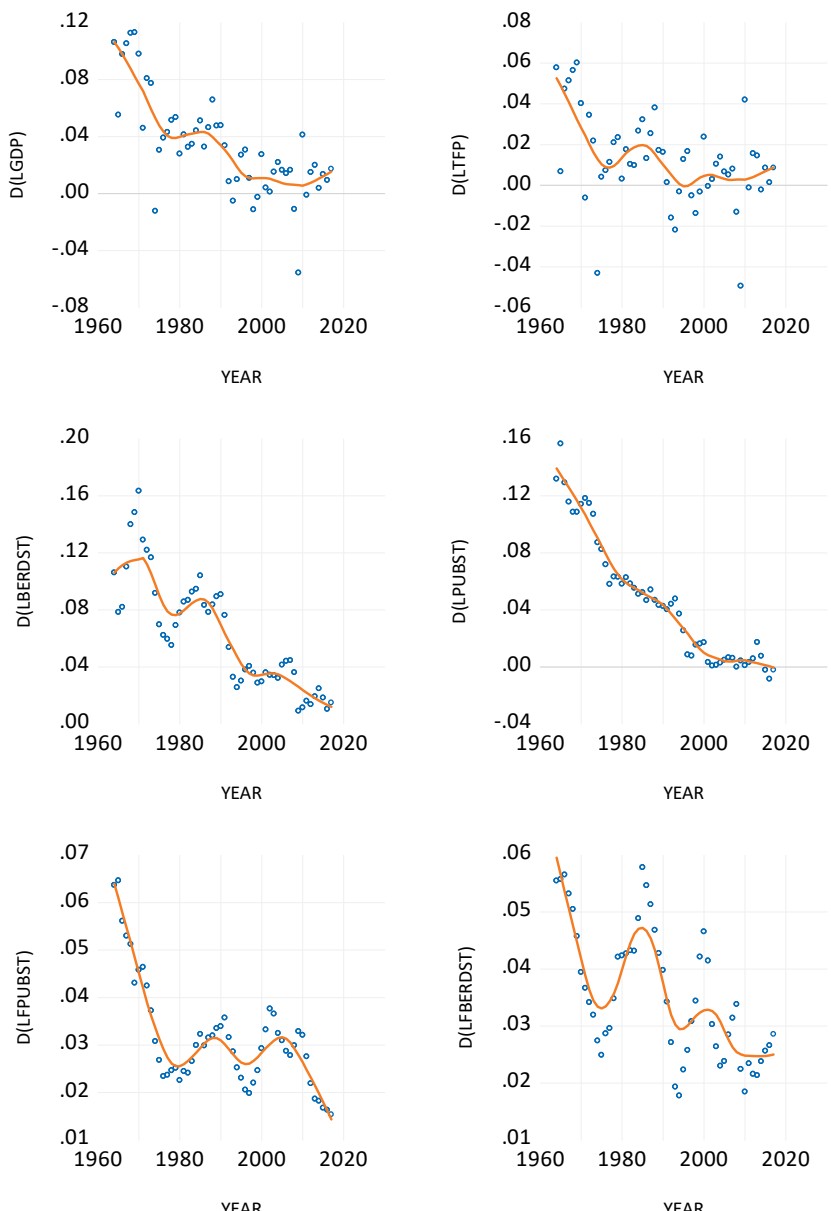

**Figure 1.** Growth rates of GDP and total factor productivity (TFP), domestic private R&D, public R&D, foreign public and private R&D for Japan in 1964–2017.

---

5   Nearest neighbor or loess fit uses 30% of the data for a weighted least squares regression and attributes the slope to the middle observation, and then shifts one observation further to repeat the procedure, resulting in a non-linear line.

## 3. Methodology

### 3.1. The Cointegrated VAR Approach

We use the regression approach for dynamic simultaneous equation estimation, in particular the cointegrated VAR or vector-error-correction (VEC) approach. This approach generates a difference equation system in the six variables mentioned above. It combines variables in levels and differences. We assume from the beginning that they are all endogenous unless testing shows the opposite. It goes beyond single equation estimation because it takes the mutual interaction between all the variables mentioned above into account in estimation, simulation and policy analysis. We start from a VAR in levels as follows:

$$y = A_1 \, y(-1) + A_2 \, y(-2) + Cx + u \tag{1}$$

All variables are dependent on the lags of all others[6]. A VECM approach may be more appropriate in the presence of unit roots and cointegration. In particular, *r* cointegrating equations, *CE*, provide information on long-term relations of variables in levels in addition to a VAR model in differences. Equation (1) can be rearranged as VECM as follows (Patterson 2000; Lütkepohl 2005; Jusélius 2006; Kilian and Lütkepohl 2017).

$$dy = \alpha\beta'y(-1) + Bdy(-1) + Fx + u \tag{2}$$

Here *dy* is the (6, 1)-vector (6 rows, 1 column) of first differences of the $K = 6$ variables introduced above. If the term $\alpha\beta'y(-1)$ were absent, we would basically have a system for the six variables in differences regressed on the lags of all six differenced variables. Estimation in differences avoids problems from variables with unit roots. $A_p$ or *B,* for (1) and (2), respectively, is the (6, 6) matrix of coefficients of the (6, 1)-vector of lagged terms with lag $p = 1, 2$, and perhaps more or less as obtained from testing for lag length. *C* and *F* are (6, 2)-matrices of the coefficients of the vector of exogenous variables $x' = (c, t)$, in the first instance only constant and time trend; here we also add unit dummies for the periods before the data revisions. $\beta$ is a (6, r)-matrix of coefficients of the log-term relations, where cointegration tests provide the number *r*; it may include a constant and a coefficient for a time trend. $CE = \beta'y(-1) = E(u(-1)) = 0$ represents the long-term relations, which may also include a time trend and a constant. $\alpha$ is the (6, r)-matrix of adjustment coefficients indicating how strongly the model reacts to disequilibrium, $u(-1) \neq 0$.

The econometric analysis of cointegrated variables leads to several possible outcomes. If the result from cointegration tests is $r = 0$, no cointegration, Equation (2) should be estimated without the term $\alpha\beta'y(-1)$; if $0 < r < K$, Equation (2) is estimated as written; if $r = K = 6$, there are no unit roots and the model can be estimated in levels as in Equation (1) (see Patterson 2000; Davidson and MacKinnon 2004). If all adjustment coefficients in $\alpha$ for one of the six equations turn out to be insignificant, that variable is called weakly exogenous because it does not react to disequilibrium deviations from the long-term relations[7]. The practical difficulty here is to find the number *r* of long-term relations. We use the trace test and the maximum eigenvalue test, together called the Johansen test. In case of $r = K - 1$ cointegrating equations, the latter are pairs of variables. Kilian and Lütkepohl (2017) suggest testing for pairwise, triple-wise, etc. cointegration in order to cross check for the Johansen tests for the whole system. The estimation method is maximum likelihood.

In the second step, we will carry out analysis of permanent policy changes by way of increasing the intercept of the equation(s) for domestic public R&D, private R&D and foreign private and public

---

[6]  Unlike other areas, we do not assume that there is another model with contemporaneous regressors behind it.
[7]  Other definitions of exogeneity deal with having also no impact on a variable *y* from other variables in the differenced part because of zero elements in *B* of Equation (2) (Patterson 2000). Then we will have an additional exogenous variable x with coefficient matrix *F* extended and the number of endogenous variables is reduced by one. The differenced equation then does not depend on any other variable and is a purely autoregressive process of a variable in differenced form if it has a unit root.

R&D by 0.005, a half percentage point, and comparing the old and the new solution. A change of a half percent as used by Coe and Helpman (1995) may be small enough to leave the model unaffected and the policy not being subject to the Lucas critique (see Kilian and Lütkepohl 2017). This can show whether or not additional public R&D generates positive effects on business R&D, TFP and GDP and what the internal rates of return are. We analyze also changes of private R&D in order to see how it affects the other variables, in particular foreign R&D variables, and changes of foreign public and private R&D stocks in order to see whether Japan obtains positive knowledge spillovers or predominantly negative competition effects. Finally, a temporary change of GDP (growth) can tell which R&D variables are pro- or counter-cyclical.

## 3.2. Net Gains and Internal Rates of Return

The benefit from policy changes of R&D and GDP variables for each year is the achieved difference of the GDP from baseline. The additional costs are 0.005 of the R&D variable in the initial year, and later the yearly additional private and public investment. The method of using intercept changes in a dynamic model results in exact numbers of yearly changes in real time. Subtracting the yearly costs from the yearly benefits yields the yearly gains. Discounting them at a conventional rate of 4% or any other rate allows adding them up and seeing whether they are positive. In addition, when the costs precede the benefits one can calculate the internal rate of return (*irr*), which is the discount rate that brings the sum of discounted present values to zero. This latter method may be problematic, however, for cases where gains go from negative to positive and then to negative and to positive again[8].

## 4. Estimation Results

### 4.1. Unit Root and Cointegration Tests

Uni-variate unit root analysis may be superfluous if all variables are I(1) and system methods can be better used instead. However, it may help seeing whether there are I(0) variables or near unit roots, with coefficients between 0.8–0.95, or unit roots with coefficients between 0.95 and unity or even higher (Enders 2015; Patterson 2000)[9]. We find near unit roots (see Appendix A. Unit Roots) suggesting to test for the number of cointegrating relations, *r*, and the number of unit roots (eigenvalues), $(K - r)$, in the system using the maximum-eigenvalue and trace tests for VECMs.

Table 1 shows that we get pairwise cointegration at the 1, 5 or 10 percent level. Therefore, we should have $K - 1$ cointegrating equations, which goes against the result for the Johansen test for the whole system, which would suggest three or four cointegrating equations (see Appendix B. Cointegration Details). The argument of low power would have to be valid in the range of 0.15−0.25 to justify rejecting the hypothesis of at most one cointegrating equation. We use the model with five cointegrating equations. Alternative models with four or three cointegrating equations are shown in Appendix B.

---

[8]    This method using the exact time resolution from a VECM for the rate of return calculation was first developed and used in the related R&D literature by Soete et al. (2020a). Sussex et al. (2016) also estimated a VECM for medical research, but they did not carry out dynamic simulations of the model with all feedback to get an internal rate of return.

[9]    The numbers give only a rough first indication. We also found a variable with $p > 0.05$ for a unit root when the coefficient was only 0.3 rather than unity in related work. See the example of Enders (2015, p. 236). We do not use a fractionally cointegrated VAR because "easy interpretation is lost for fractionally differenced variables... reliable estimation of fractionally integrated processes requires larger samples than typically available in macroeconomics"; "how well this approach works in small samples is not known". (Kilian and Lütkepohl (2017), pp. 23 and 99).

**Table 1.** Pairwise Johansen cointegration tests, *p*-values.

| Test (a) | Trace Test | Trace Test | Max-Eigenvalue | Max-Eigenvalue |
|---|---|---|---|---|
| Hypothesis | No coint. equation | At most 1 | No coint. equation | At most 1 |
| TFP, BERDST | 0.0017 | 0.1803 | 0.0028 | 0.1803 |
| BERDST, PUBST | 0.0001 | 0.2415 | 0.0001 | 0.2415 |
| PUBST, FPUBST (b) | 0.0013 | 0.2459 | 0.0014 | 0.2459 |
| FPUBST, GDP | 0.0074 | 0.1770 | 0.0140 | 0.1770 |
| GDP, FBERDST | 0.0022 | 0.1686 | 0.0040 | 0.1686 |

(a) Based on VECM 1967–2017 with three lags and including dum96, dum08, dum13. (b) Based on VECM 1967–2017 with two lags and the dummies.

### 4.2. The Model with Five Cointegrating Equations

The model is stable also after imposing constraints on the less significant adjustment conditions. The long-term relations with t-values in brackets are as follows.

$$\text{CE1} = E(u_1(-1)) = 0 = \text{LTFP}(-1) - 0.257\text{LBERDST}(-1) + 0.0032t + 3.21 \tag{3}$$
$$[-23.44] \qquad [3.2]$$

$$\text{CE2} = E(u_2(-1)) = 0 = \text{LBERDST}(-1) - 0.88\text{LPUBST}(-1) - 0.0248t - 1.34 \tag{4}$$
$$[-32.16] \qquad [-5.98]$$

$$\text{CE3} = E(u_3(-1)) = 0 = \text{LFPUBST}(-1) + 0.16\text{LPUBST}(-1) - 0.0256t - 11.8 \tag{5}$$
$$[16.75] \qquad [-9.2]$$

$$\text{CE4} = E(u_4(-1)) = 0 = \text{LFPUBST}(-1) + 0.1666\text{LGDP}(-1) - 0.0267t - 12.93 \tag{6}$$
$$[16.22] \qquad [-9.71]$$

$$\text{CE5} = E(u_5(-1)) = 0 = \text{LFBERDST}(-1) - 0.37\text{LGDP}(-1) - 0.0266t - 5.59 \tag{7}$$
$$[28.2] \qquad [-7.42]$$

An alternative way of writing these equations would be to put all but the first term from the right-hand side to the left with the due change in sign. The cointegrating equations show important results. The basic mechanism of thinking about R&D works well for Japan: Domestic public R&D, LPUBST, has a positive impact on private R&D, LBERDST, in Equation (4), which in turn has a positive partial long-term impact on LTFP in Equation (3)[10]; the response of foreign countries to higher Japanese growth is to increase private R&D and to reduce public R&D, according to Equations (6) and (7). The causality may go both ways in the long-term relations and in the analysis of foreign policy changes in Section 5.2. Japanese public R&D leads to a reduction of foreign public R&D and vice versa. LTFP has a positive partial long-term effect on LGDP through the whole system. All variables are de-trended by the time trend t (see Wooldridge 2013, chapter 10.5). Exogenous time trends in the long-term relation and constants in the equations for the growth rates make sure that all variables in levels grow in the stable steady state. However, five long-term relations can determine only five variables conditional on the sixth, and therefore the whole system determines ultimately all the variables for each point in time. Each of these long-term results is only a partial effect.

The common belief that foreign R&D capital stocks have stimulated Japan's R&D (Luintel and Khan 2004) is supported here through the long-term effect of foreign private R&D stocks on the GDP in (7), which in turn decreases foreign public R&D leading to higher domestic public and private R&D

---

[10] We do not use a theoretical model here, but (3) and (4) can be related to the marginal products of TFP with respect to BERDST and PUBST for a generalization of the CES function for TFP generation. See Section 5.4.

and TFP. As this interpretation takes the long-term relations at face value, the reverse causality effect of the foreign reaction to Japanese growth mentioned before is not dominant in this series of partial interpretations. Analysis of policy changes below will reveal more information on the interaction of the six variables below.

The complete model for the estimation period 1967–2017, abbreviating the long-term relations as CE1–5 as in Equations (3) to (7), is as follows in Equations (8)–(13) below (t-values in parentheses for adjustment coefficients). We use as abbreviation *A* for *LTFP*, *B* for *LBERDST*, *P* for *LPUBST*, *Y* for *LGDP*, '*' for the corresponding foreign variables and *D* for the first difference with respect to time. " … " indicates terms in first differences, constants and dummies, presented in Appendix C 'First-difference terms as equilibrium version of the estimated VECM'.

$$D(A) = 1.4CE1 - 0.15CE2 + 10.31CE3 - 10.41CE4 + 1.63CE5$$
$$[2.89] \quad [-2.92] \quad [4.79] \quad [-4.75] \quad [5.63]$$
(8)

$$D(B) = 0.65CE1 - 0.158CE2 + 4.78CE3 - 4.768CE4 + 0.49CE5$$
$$[1.77] \quad [-1.4] \quad [3.34] \quad [-3.2] \quad [2.24]$$
(9)

$$D(P) = -0.48CE1 + 0CE2 - 2.6CE3 + 2.486CE4 - 0.11CE5$$
$$[-2.53] \quad [NA] \quad [-3.04] \quad [2.86] \quad [-0.97]$$
(10)

$$D(P^*) = 0CE1 + 0CE2 - 0.0216CE3 + 0CE4 + 0.11CE5$$
$$[NA] \quad [NA] \quad [-1.274] \quad [NA] \quad [2.14]$$
(11)

$$D(Y) = 2.44CE1 + 0CE2 + 13.85CE3 - 14.18CE4 + 1.89CE5$$
$$[4.5] \quad [NA] \quad [5.66] \quad [-5.72] \quad [5.9]$$
(12)

$$D(B^*) = 0.377CE1 + 0.156CE2 + 2.31CE3 - 2.55CE4 - 0.21CE5$$
$$[2.42] \quad [3.02] \quad [3.88] \quad [-4.05] \quad [-2.33]$$
(13)

Adjusted R-squared for Equations (8)–(13) are 0.72, 0.97, 0.986, 0.935, 0.85, 0.936[11]. When estimating a vector-error-correction model, we set adjustment coefficients with t << 1 to zero. The lowest t-value for adjustment coefficients is the last term in Equation (10). When going to stricter t-values, the p-value for the chi-square test on all restrictions decreases strongly; this may be legitimate in the vein of testing theories, but the model then moves away from 'letting the data speak.' The t-values for adjustment coefficients shown in Equations (8)–(13) below suggest that we are not very restrictive in setting coefficients to zero. The coefficient with the lowest t-value is not small, however, and putting the adjustment coefficient to zero causes a large drop in the $p = 0.966$ of the chi-square statistic testing the constraints.

### 4.3. Relations with the Literature on Japan's Growth

Figures 2 and 3 below include the baseline scenario and the data, showing that TFP has about three phases of constant growth rates, until 1974, 1975–1991, 1992 to present, where earlier phases have higher growth rates, and in the last two phases growth rates are below those mentioned in the beginning of the introduction. Foreign R&D variables have about constant growth rates, public R&D has zero growth in the second half of the total period, and private R&D has decreasing growth rates. Hayashi and Prescott (2002) and Kaihatsu and Kurozumi (2014) interpret the trend in TFP (not corrected for human capital) as exogenous. However, here TFP is endogenous as adjustment coefficients are statistically significant in Equation (8). Other authors point to effects of financial problems affecting GDP. We will look at this argument in Section 5.3 in the form of a temporary change of GDP growth.

---

[11]　See Appendix C for other statistics.

For the period after 1990, Bottazzi and Peri (2007) point out that R&D employment is stagnant in Japan. This goes together in our data with decreasing growth rates of domestic private and zero growth of domestic public R&D stocks. Branstetter and Nakamura (2003) point out that private R&D expenditure flows stagnated or grew slowly during the 1990s leading to mostly decreasing growth rates of private (and public) R&D stocks in our data and simulations below[12].

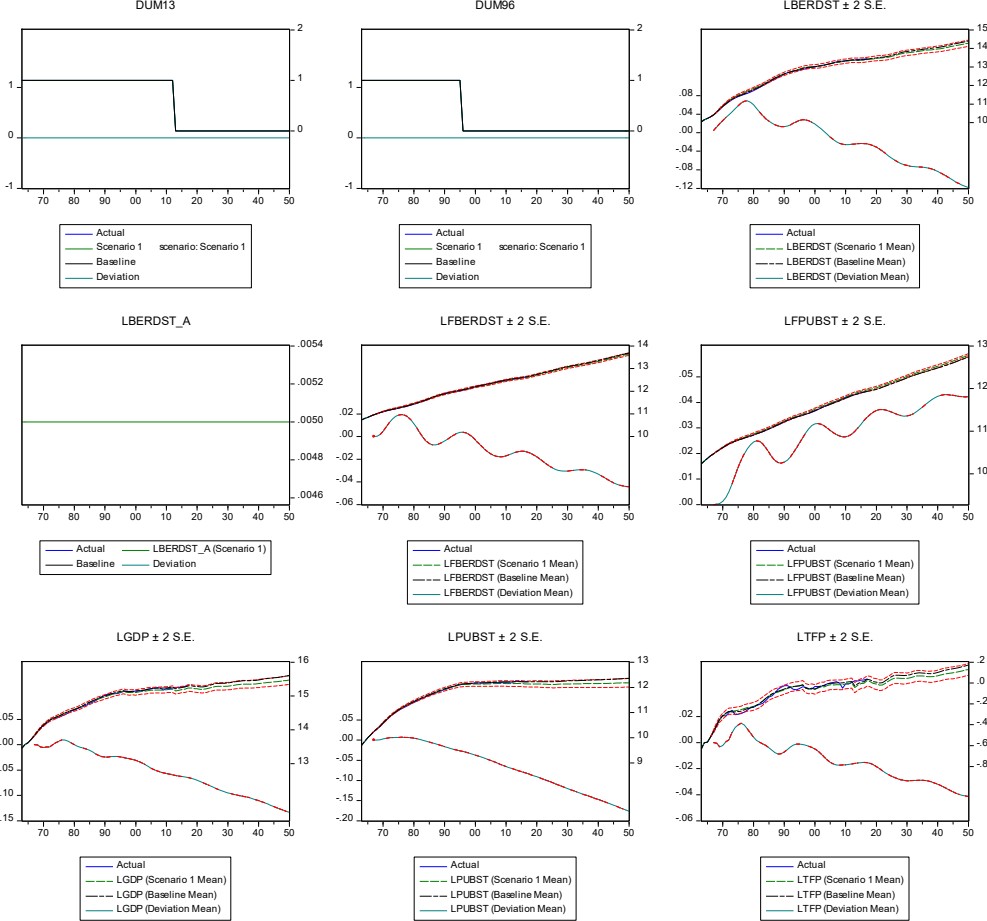

**Figure 2.** Effects of a permanent policy change on private R&D stock until 2050 from the VECM with five long-term relations. The left axes measure differences to baseline at the lower curve. The right-hand axes measure levels of the scenario, baseline and actuals at the higher set of curves in each graph. Confidence intervals are for the policy scenario. Their shift is roughly the same as that from baseline and therefore the lines are on each other.

---

12  Miyagawa and Ishikawa (2019) disaggregate the macro variables according to sectors and find decreasing R&D efficiency in some sectors.

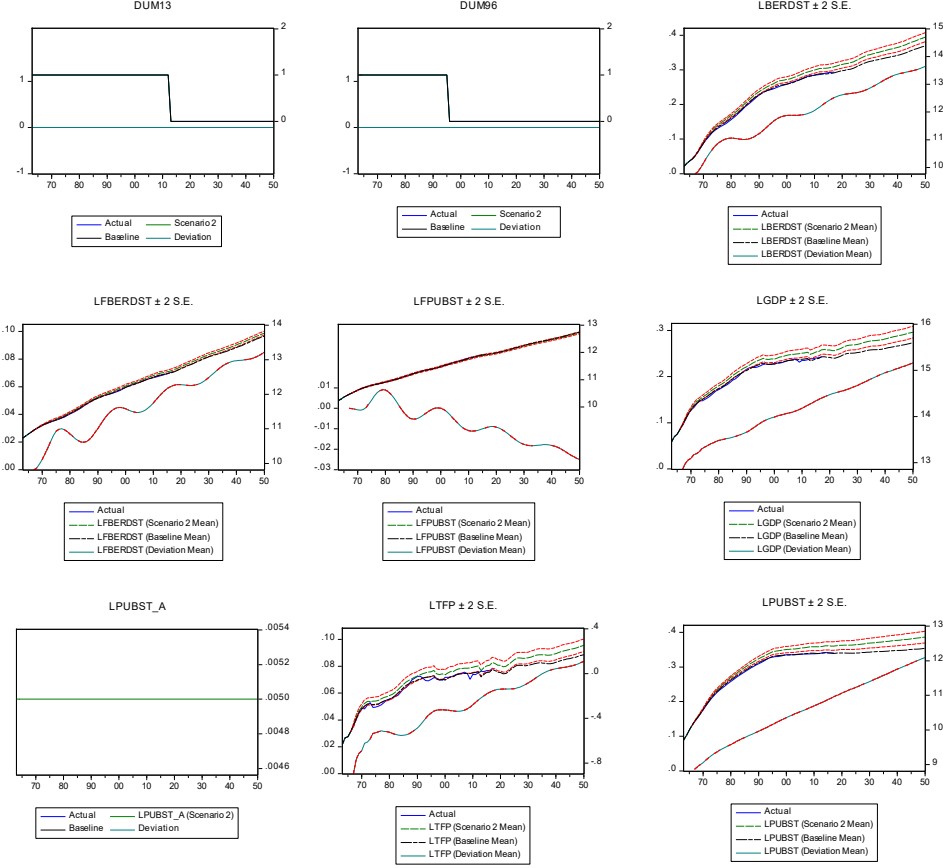

**Figure 3.** Effects of a permanent policy change on public R&D stock until 2050 from the VECM with five long-term relations. Axes and confidence intervals as in Figure 2.

The critical question then is what is behind the slow growth that tears growth of R&D and TFP down. It is beyond the scope of this paper to give a full explanation but some of the relevant reasons may be the following:

(i)　Branstetter and Nakamura (2003) defend stagnant human capital supply as a cause of stagnant R&D employment for PhD-level engineers, and Arora et al. (2013) for IT software and Kim (2016) for durable manufactured goods. Arora et al. (2013) argue that Japanese immigration practices are restrictive, offshoring is not used much, and domestic graduates are scarce. De Loo and Ziesemer (2000) mentions the low number of PhD grants in Japan. Consequently, Japanese firms do software skill-intensive research in the USA and patent there. Overall, stagnant R&D employment caused by a limit in human capital supply is an interesting explanation; R&D expenditures then only drive up human capital prices.

(ii)　Kim (2016) finds a negative though insignificant sign of private R&D on TFP on the firm level in Japanese manufacturing and argues that R&D expenditure may go to foreign subsidiaries and thereby reduce the productivity of headquarters.

(iii)　The number of researchers as a share of the population may be too high in Japan (see Goto 2000, Table 1), leaving too few high skilled workers for production. If many are doing research, there may be an insufficient number of scientists and engineers to cooperate in development and production; a given accounting number at the macro level may go together with an imbalance between the number of researchers and applying engineers (see Grossman 1989 for the theoretical underpinning). The Japanese tax system may support such an imbalance because it allows deducting changes rather than levels in R&D expenditure from the sum of taxes paid (Goto 2000). The incentive then is to go beyond optimum R&D expenditures. This may be a

reason why Japanese firms have possibly strengthened research at the cost of productivity in production. The literature studied so far for this paper shows that R&D in Japan is strong in every type of input, but it does not look at engineers in production. Moreover, if all researchers are fully employed, additional R&D expenditure may increase wages more than hiring, which increases GDP but makes TFP-enhancing projects less profitable.

(iv)　Shifts of sector shares to low productivity sectors may be an important element. Interpreting macro TFP as a weighted sum of sectoral TFPs, sectors with below average productivity possibly gain in shares of GDP in the period with negative growth rates when old industries are defended against the market trends and exiting firms are more productive than staying firms (Fukao and Kwon 2004). An example is government subsidies for specific technologies such as a smelting process (see Peck et al. 1987), which goes beyond technological neutrality. The negative national spillovers found by Bernstein and Yan (1996) and the decreasing spillovers from large to smaller firms found by Fukao (2013) may be part of this. In recent years, the fall of Japanese sectors behind their US competitors (Branstetter and Nakamura 2003) may have continued this effect. Fukao and Kwon (2004) state that "while TFP growth decelerated in many manufacturing industries, it picked up in non-manufacturing industries"[13]; the strongest counterpart in the rise of services as a share of GDP is the fall of non-manufacturing industries, implying that sectors with stronger TFP growth are shrinking relatively more.

(v)　Falling capacity utilization and exit of efficient firms may have contributed to this (Jones and Yokoyama 2006).

All explanations seem to be complementary and in line with our R&D stock data and our model. The timing fits the low growth rates of GDP mentioned in the introduction. Finding the exact contribution of the most important causes for the decline of TFP is a topic going beyond the scope of this paper.

## 5. Simulation Results

### 5.1. Results for Permanent Private and Public R&D Policy Changes

The results described so far only indicate regression coefficients but do not include indirect effects of all variables influencing each other mutually. In particular, variables that have no direct impact can have indirect impacts via other variables and dependent variables may have feedback effects.

To see this explicitly, we impose a change of 0.005 on the constant of the private investment Equation (9) (DLBERDST) in scenario 1 and on the constant of the public R&D stock (10) (DLPUBST) in scenario 2 in the VECM with five cointegrating equations[14]. The figures show the results for these changes. Figure 2 shows that a permanent policy change of 0.005 on private R&D has a positive effect on publicly performed R&D only for the first 16 years (until 1982) in the VECM. TFP and GDP react first negatively and then positively until 1982 and 1980, respectively, and the effect remains negative in the long run. Furthermore, foreign R&D variables react strongly, permanently only for foreign public R&D and for 36 years for foreign private R&D.

Figure 3 shows that after the policy change of public investment, public R&D stock keeps growing beyond baseline up to 32 percent. Private R&D, LBERDST, shows a similar pattern going 31 percent beyond baseline. TFP increases by eight percent and GDP by more than 23 percent; both have slightly increased growth rates (slopes of the log-level curves) indicating that the VECM is an endogenous growth model. Higher TFP also attracts foreign capital or keeps domestic capital from moving out.

---

[13]　Most studies discussed in Fukao and Kwon (2006) seem to show the opposite result, however.
[14]　See Lau (1997), Equation (22), for a formalization of permanent changes in VARs- and VECMs-related exogenous and endogenous growth theory.

Foreign private R&D goes up, as does foreign public R&D for 13 years, after five negative years; Japan creates a strategic foreign reaction to higher domestic R&D investment. As Bernstein and Mohnen (1998) find no spillovers from Japan to the USA (for an earlier period, however) and Luintel and Khan (2004) even negative ones, it is likely that Japan generates spillovers to or stimulates expenditures of the EU or other countries. This is in line with positive effects found by Luintel and Khan (2004) on other countries than the USA. However, here it comes explicitly from public R&D policy changes, which is in line with the finding of Luintel and Khan (2004) that business R&D has weaker effects than total R&D.

Table 2 presents the average values of the differences for scenarios and baseline for the years 1967–2050 for the plots of Figures 2 and 3 for the VAR in levels and Figures 2 and 3 regarding the VECM with five cointegrating equations, all evaluated at expected values. The higher effect on TFP and GDP comes from a policy change for public compared to private R&D, with effects on GDP stronger than for TFP. Foreign private R&D has on average and in the main phases of both figures the same sign as domestic private R&D, suggesting that they are strategic complements.

**Table 2.** Effects from permanent policy changes for private and public R&D: Percentage difference from baseline averaged from 1967–2050.

| Effects → (a) from Policy ↓ | Domestic Private R&D | Domestic Public R&D | Foreign Public R&D | Foreign Private R&D | TFP | GDP |
|---|---|---|---|---|---|---|
| Private R&D (VECM) | −0.019 | −0.067 | 0.028 | −0.0133 | −0.0146 | −0.054 |
| public R&D (VECM) | 0.18 | 0.18 | −0.008 | 0.049 | 0.052 | 0.129 |

(a) Negative average effects often begin with positive effects; see Figures 2 and 3.

Effects on TFP in Table 2 are strongly different on average. Results for the policy change for private R&D on TFP in the VECM in the first line of Table 2 depend on taking the averages until 2050 because they are positive in the first phase and negative in the long last phase in Figure 2, which plays a minor role under discounting. In these cases, visual inspection of the figures is more instructive than Table 2. When effects become negative policies could be stopped. Then all variables would go back to baseline and average effects of policy changes would appear to be much more positive than in Table 2.

When effects on TFP are positive, they may still not justify the additional costs in terms of public and private R&D. This suggests comparing benefits and costs. Results may look better when limiting the evaluation, in a second step carried out in Table 3, to periods of positive net gains from the policies shown in Figures 2 and 3, assuming that projects are stopped when only periods with negative effects would follow. Table 3 presents the years of positive gains, the results from calculating the gains (difference of policy to baseline of GDP minus change of cost flows for private and public R&D) as share of GDP, their sum of present values discounted at 4%, and internal rates of return.

**Table 3.** Net gains, discounted present value (DPV), internal rates of return (IRR) to additional private and public R&D.

| Effects → of Permanent Policy Change on ↓ | Years of Gains; Average Gain/GDP (a) | Sum DPV (4%) in Bill. $ | Internal Rate of Return (Payback Period) (b) | Remarks |
|---|---|---|---|---|
| private R&D (VECM) | 1972–1982; 0.004 | 95.445 | 43.66% (16) | (c) |
| public R&D (VECM) | 1968–2050; 0.044 | 2483 | 425.5% (4) | (d) |

(a) Only the yearly percentage share of TFP change multiplied by the GDP is counted as benefit. (b) Payback period is the number of years that it takes to have positive sum of gains after discounting at the internal rate of return. Period 1 is the first shock period 1967. (c) Initial costs (1967) 0.5% of private R&D stock. (d) Initial costs (1967) 0.5% of public R&D stock.

The first row shows results for the permanent policy change on private R&D compared to baseline for the VECM. The second row does this for the shock on public R&D[15]. These are financial results comparing the policy shocks relative to baseline. Public R&D shocks have long-lasting gains until the horizon of 2050 after only one year of initial losses. In contrast, private R&D shocks have a payback period of 16 years when only TFP gains are considered. The sum of present values discounted at 4% interest over all years has the same ranking: larger for the public then the private shock. The internal rate of return is also much higher for public than for private shocks on R&D. The rates of return for private and public R&D are higher than the marginal products of total R&D in the literature, which in turn was doing better than business R&D (Luintel and Khan 2004). The reason is that the traditional approach turns elasticities into a marginal product in an atemporal way or as a steady-state solution of an error correction model. In our approach of calculating net gains per year and discounting them with the internal rate of return, it also plays a role whether the benefits, costs and gains come about early or late. The high internal rates of return suggest that the gains are obtained early as indicated by the short payback period for public R&D shocks. Short but early periods of gains lead to high internal rates in the case of public R&D, and long payback periods with late gains lead to relatively low internal rates of return. Payback periods after discounting at the internal rate of return are inversely proportional to internal rates of return as indicated in the second-to-last column of Table 2.

Overall, the limited effects of additional private R&D indicate that private R&D is close to optimal given its risks. There are a couple of reasons why the rates of return are so high and payback periods so low for public R&D. First, we do not consider the additional costs for capital and labor in production of the higher GDP; it is exactly the purpose of growth policies to increase employment and wages and attract international capital. We do not include these indirect effects in the costs, because they are not only costs for firms but also income for households, and therefore are canceled out from a welfare perspective. Second, the analysis is ex-post, whereas decisions are taken under uncertainty and risk; implicit risk premia may be high here. Third, a log–log specification as used here has decreasing marginal products in case of positive coefficients; by implication, rates of return may be higher if less is done in terms of inputs as indicated by the relatively stronger expansion of private compared to public R&D in Japan as documented by Archibugi and Filippetti (2018).[16] Fourth, policies affect international R&D, which generates spillover repercussions (knowledge externalities or complementary strategic reactions) to the economy under consideration, which are predominantly positive. As a policy against slow growth, additional public R&D has a stronger effect than additional private R&D.

Table 2 and Figures 2 and 3 indicate that foreign private and public R&D stock are endogenous for Japan, whereas Luintel and Khan (2004) find only a 9% probability for this when foreign R&D stocks are built from the sum of private and public flows rather than having two foreign stocks, for private and public R&D separately. Defining foreign public and private R&D stocks separately in our paper makes the impact of Japan's innovation on foreign R&D visible.

*5.2. Results for Changes of Foreign Private and Public R&D*

As Luintel and Khan (2004), working with a similar method but leaning on long-term relations only, did not get clear results regarding foreign total R&D spillovers it may be worth investigating the question again, separately for foreign private and public spillovers, using results from permanent policy changes. Figure 4 shows the effects of a shock on foreign private R&D stocks. All variables except for domestic public R&D go up for some years. The reaction of TFP is smaller than the reactions

---

[15] We often use the word "shock" for intercept changes rather than for residuals as the SVAR (structural vector-autoregression) literature is doing (Kilian and Lütkepohl 2017).

[16] Under the opposite assumption of equal marginal products of private and public R&D in TFP production, it is possible to obtain the parameters of a VES production function from the VECM estimates (see Section 5.4 "VES parameters from VECM estimates").

of the foreign R&D variables. Domestic private R&D reacts even less. Foreign private R&D has a clearly positive but small and transitional effect on Japan's R&D, TFP and GDP.

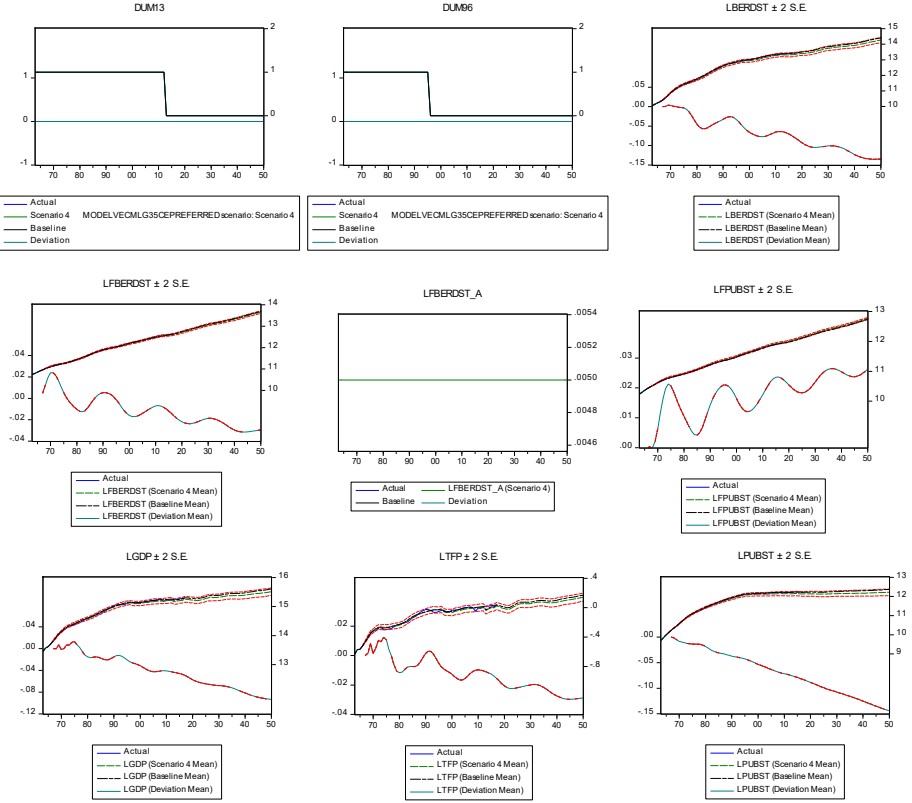

**Figure 4.** Effects of a shift of foreign private R&D stocks from the VECM with five long-term relations. Axes and confidence intervals as in Figure 2.

Figure 5 shows the effect of a shock on foreign public R&D. Foreign and domestic private R&D decrease strongly. Domestic public R&D, TFP and GDP fall, also in terms of growth rates. This result for foreign public R&D is the opposite of that for foreign private R&D. The reason lies in the opposite reaction of foreign private and public R&D. When foreign public R&D goes up, foreign private R&D goes down, and so does domestic private R&D, pulling down TFP and GDP. This is an example for the more general lesson that co-movements depend on the source of the shock.

In sum, foreign private R&D shocks stimulate domestic private R&D, TFP and GDP but not the other variables. Foreign public R&D shocks do the opposite. Having foreign R&D stocks separately in the data for private and public R&D variables, permanent policy changes to them clearly play a role for private and public R&D in Japan and therefore it is plausible that aggregating foreign private and public R&D yields unclear results in Luintel and Khan (2004). Together with the results from the previous section, showing the impact of shocks in Japan on the foreign stock variables, this implies that Japan's R&D is well integrated into the world economy; effects go both ways.

### 5.3. Checking Cyclically of Public R&D: The Effects of a Temporary GDP Change

A temporary change of GDP (growth) in 1967 of 0.005, a half percentage point, shown in Figure 6, generates a negative reaction of GDP in the next three years and then a positive one for four years and the opposite for public R&D, which is in line with the counter-cyclical result for Japan in Pellen et al. (2017) and the negative partial lagged short-run effect of our VECM (see Appendices A and C). In contrast, private R&D and TFP are pro-cyclical. Foreign private and public R&D follow Japan's GDP quickly as a policy response. While all other effects phase out, GDP and public R&D have a small positive

long-run effect in the VECM. TFP does not react to temporary changes of GDP in the longer run; GDP and TFP both return to baseline in the long run.

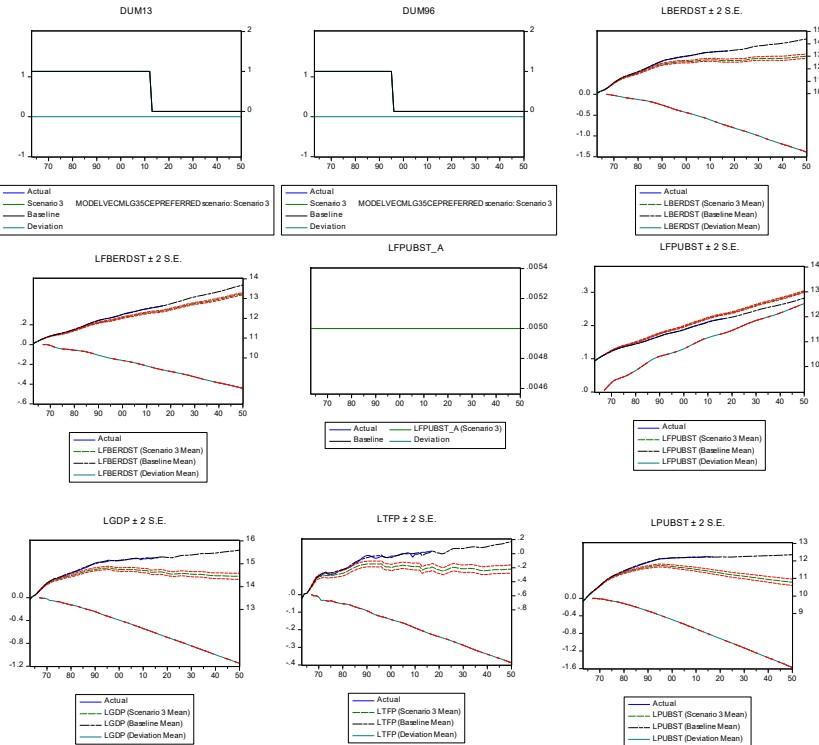

**Figure 5.** Effects of an increase in foreign public R&D stocks from the VECM with five long-term relations. Axes and confidence intervals as in Figure 2.

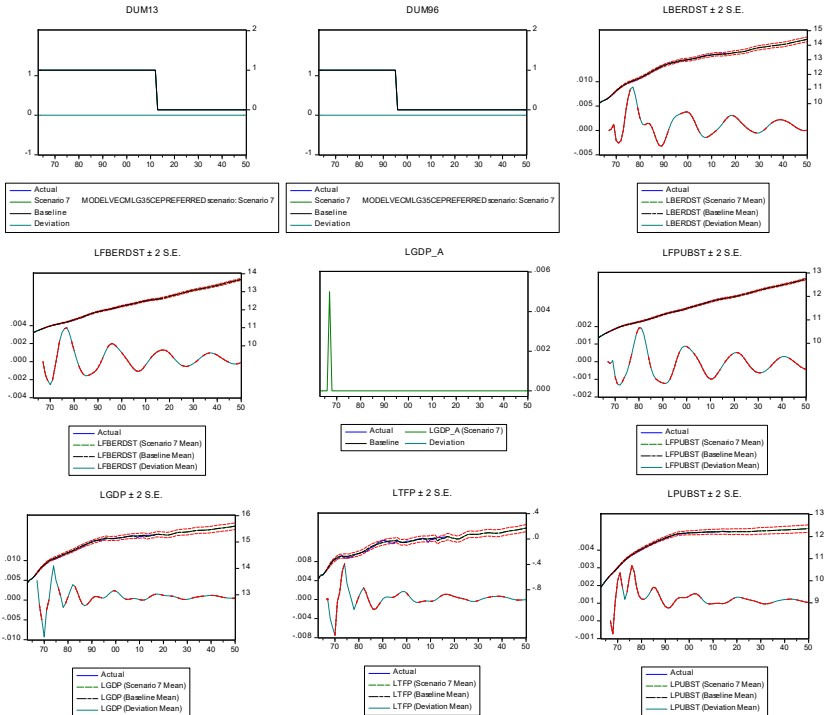

**Figure 6.** Effects of a temporary change of GDP (growth) from the VECM with five long-term relations. Axes and confidence intervals as in Figure 2.

*5.4. VES Parameters from VECM Estimates*

Ziesemer (2020) provides an endogenous growth model with the variables of the VECM using Mukerji (1963) VES function for TFP production with *A* for TFP, $R_b$ for private R&D stock and $R_g$ for public R&D stock, and the corresponding foreign variables marked with '* '.

$$A = \left[ a(A_{-1})^{\varphi} + h e^{\alpha b t} R_b^{\alpha} + (1-h) e^{\beta x t} R_g^{\beta} + d R_b^{*\gamma} + f R_g^{*\mu} \right]^{\frac{1}{\epsilon}} C$$

If $\alpha = \beta = \epsilon = \gamma = \mu$ we would have a CES (constant elasticity of substitution) function. If they were all zero, we would have a Cobb–Douglas function. From the formulas for marginal products $F_2$ and $F_3$, for private and public R&D we get

$$logA = \frac{(1-\alpha)}{(1-\epsilon)} logR_b - \frac{\alpha b}{(1-\epsilon)} t + \frac{1}{(1-\epsilon)} \log\left( \frac{\epsilon F_2}{h\alpha C^{\epsilon}} \right)$$

$$logA = \frac{(1-\beta)}{(1-\epsilon)} logR_g - \frac{\beta x}{(1-\epsilon)} t + \frac{1}{(1-\epsilon)} \log\left( \frac{\epsilon F_3}{(1-h)\beta C^{\epsilon}} \right)$$

Equating the left-hand sides, we get

$$logR_b = \frac{(1-\beta)}{(1-\alpha)} logR_g + \frac{(\alpha b - \beta x)}{(1-\alpha)} t + \frac{1}{(1-\alpha)} log\frac{F_3}{F_2} + \frac{1}{(1-\alpha)} log\frac{h\alpha}{(1-h)\beta}$$

The first of these three theoretical relations correspond to the empirical long-term relation (3) in the VECM estimate, and the third corresponds to (4). This allows backing out the parameter values for the special case $b = x$, equal factor augmenting rates of technical change in the TFP production function, as $b = x = -0.208$, $\alpha = -0.064$, $\beta = 0.063$, $\epsilon = -3.1385$. Under the additional assumption $F_2/C^{\epsilon} = F_3/C^{\epsilon} = \phi$, we get from the constants in (3) and (4): $h = 1.3223$, $\phi = 4.5582 \times 10^{-8}$. The latter conditions impose the efficiency conditions of equal marginal products for private and public R&D from the growth model. Using these values in the TFP production function above shows that private and public R&D have a positive effect on TFP, because for each of them two of the relevant parameters are negative. The elasticities can be read off directly from the VECM conditions (3) and (4). Our interpretation of these results is that the relation between TFP and R&D stocks is not a CES, let alone a Cobb–Douglas function. Other VES functions like that of Revankar (1971), even with disembodied technical change (De Loo and Ziesemer 2000), do not give us log-linear marginal products (Thach 2020), which could be linked to the long-term relations of a VECM.

## 6. Summary and Conclusions

Our VECM analysis has shown that public R&D affects business R&D, TFP, GDP and foreign private R&D positively, in the short and the long run, and foreign public R&D for thirteen years. Papers assuming panel homogeneity find no effects of public R&D (Van Elk et al. 2019; Herzer 2020). Our models with three or four cointegrating equation (not shown) have stronger effects of public R&D changes than our preferred model with five cointegrating equations. Therefore, it is of utmost importance to be careful about model selection by way of revealing the weaknesses of all models as we have done with the information in Table A3, which favors five cointegrating equations. Models with panel homogeneity underestimate the gains from additional public R&D; models with fewer than five cointegrating equations overestimate the gains from additional public R&D[17].

---

[17] An asymmetric VAR with up to four lags (not shown) shows slightly lower effects than our preferred model, but the trace test would require accepting a *p*-value of 40 percent for the hypothesis of "at most five cointegrating equations" (Table A2).

Changes of private R&D have positive but much weaker effects on TFP (than changes of public R&D) with an internal rate of return that is still reasonably high. Japan's slow growth could be speeded up mainly through more public R&D stimulating private R&D. Possibly, this could strengthen within-sector TFP growth in close interaction with foreign R&D, which was the major driving force of the high-speed growth period 1955–1970 (Fukao et al. 2019).

Changes in foreign private R&D have a positive impact on Japan's growth. Changes of foreign public R&D have strongly negative effects of Japan's growth because Japan's public R&D reacts positively and reduces Japans private R&D and TFP. Japan's public and private R&D then should react to work against these negative foreign impacts if they happen to occur.

The net gains from additional public R&D are strongly positive with high internal rates of return. Our recommendation as to "what policy changes would allow productivity to grow again" (Hayashi and Prescott 2002) has two logical steps. First, as long as the causes for low TFP growth discussed in the literature continue to exist, we suggest a special form of an R&D policy. Public R&D changes should lead the way and stimulate private R&D, because the effect on TFP is larger than that of a change of public R&D as can be seen from comparing Figures 1 and 2, where TFP growth rates, the slopes of LTFP, are larger after a permanent change of the intercept as in endogenous growth theory. The answers to the research questions imply that public and private R&D should develop in a more balanced way, as they do after a shock to public R&D but not after a shock to private R&D. Public and private R&D are more complementary when firms do more basic R&D according to the discriminant analysis of Soete et al. (2020b), and private basic R&D has been increased for Japan (Archibugi and Filippetti 2018). Keeping them more in balance turns out to be good for TFP growth in our analysis. However, Ziesemer (2019) finds that private R&D shocks may have stronger effects than public R&D shocks when including mission-oriented R&D[18]. Together these results might suggest that private and public R&D go up together in a coordinated way. Second, policy should improve upon the lack of human capital discussed in the literature, if possible. The role of mission-oriented R&D may be worth explicit consideration (Mazzucato 2018) but this requires a different paper.

Additional private and public R&D of Japan also have positive effects on foreign private and public R&D. Thus, it is hard to sustain that Japanese R&D has no effects on other countries' R&D decisions. In addition, we have shown that changes of foreign public R&D have negative effects on Japanese private and public R&D, TFP and GDP. Effects of shocks on foreign private on domestic private and public R&D are negative in the VECM.

Backing out the VES parameters for a TFP production function indicates that the latter are unlikely to be of the CES type or even Cobb–Douglas, but are closely related to a theoretical model.

**Funding:** This research received no external funding.

**Acknowledgments:** I gratefully acknowledge useful comments from two anonymous reviewers, Hugo Hollanders, Georg Licht, Luc Soete, Bart Verspagen and participants of two meetings at EU DG R&I. Responsibility is entirely mine.

**Conflicts of Interest:** The author declares no conflict of interest.

## Appendix A. Unit Roots

The standard tests for unit roots provide contradictory results in our data set. We use the DF-GLS test without breakpoint in Table A1, because it is an efficient test and Hjalmarsson and Österholm (2010) use it for comparison with the local-to-unity model. For the DF-GLS test, all *p*-values show that deviations from zero are insignificant and we have unit roots for all variables (in natural logarithms here and in all other uses below), provided there are no breakpoints. However, the Zivot–Andrews test indicates break years 1987 or 1988 for trend and intercept, and all t-values are too low to reject

---

[18] A companion paper includes GBA(O)RD, government appropriations for mission-oriented social objectives implying availability of data for fewer periods.

the hypothesis of unit roots. As we do not get the coefficients for the lagged dependent variables, we cannot see whether variables are stationary or have near unit roots. Therefore, we use also the univariate Vogelsang–Perron ADF unit root test with break point. We use both the innovational outlier test and the additive outlier test. Table A1 again shows the results. We use the Akaike information criterion for the lag length as for the DF-GLS test, because this is also suggested for the lag length test for the VECM (Kilian and Lütkepohl 2017). In the first step we test for break in intercept and trend using the F-test for a joint break. If one of them is insignificant we use the Dickey–Fuller min-t test. We successively drop trend break, intercept break, trend or intercept whenever they are insignificant. Table A1 shows coefficients between 0.5 and 0.97 and high probabilities for a unit root for all variables. All variables have a unit root with probability 0.2 or higher in at least one of the tests. If we look for all variables at the test with the higher *p*-value for a unit root, then the coefficients are between 0.75 and 0.97 instead of unity. This means we do not have exact unit roots or stationary variables but rather near unit roots. Break dates are very heterogeneous in Table A1 for the ADF breakpoint tests. The results should be used with the utmost caution, however, as the tests are univariate and therefore do not include any regressors on the right-hand side except its own lags. This assumption is called (weakly) exogenous in other contexts. The tests, therefore, are not valid for endogenous variables. Fukao and Kwon (2006) therefore excludes them from her book because they can be very misleading and suggests using only systems methods leaning on VECMs. We include them only because Hjalmarsson and Österholm suggest that they give an indication of the order of integration or the presence of near unit roots.

**Table A1.** Unit root tests without and with break point (a).

| Variable | LGDP | LTFP | LBERDST | LPUBST | LFBERDST | LFPUBST |
|---|---|---|---|---|---|---|
| Dickey–Fuller GLS without breakpoints. Dependent Variable: D(GLSRESIDUAL); coefficient of lagged residual. | | | | | | |
| Coefficient | 0.019 | −0.035 | −0.008 | −0.0056 | −0.0099 | −0.0068 |
| t-value (*p*-value) | −1.246 (0.219) | −1.011 (0.3167) | −1.36 (0.1796) | −1.61 (0.113) | −1.001 (0.32) | −0.875 (0.3857) |
| Zivot–Andrews unit-root test with breakpoint determination for intercept and trend. | | | | | | |
| t-value (c) unit root | −3.454 yes | −3.4478 yes | −4.393 yes | −2.45 yes | −4.2876 yes | −3.3235 yes |
| Break year | 1988 | 1987 | 1988 | 1987 | 1984 | 2008 |
| ADF unit-root test with break | | | | | | |
| Coeff. (b) | 0.85 0.81 | 0.51 0.75 | 0.88 0.92 | 0.952 0.93 | 0.72 0.91 | 0.90 0.965 |
| *p*-value (c) | 0.75 0.19 | 0.0155 0.47 | 0.0194 0.2 | 0.45 0.21 | 0.21 0.70 | 0.30 0.99 |
| Break date (d) | 1992 1986 | 1992 1992 | 1997 1997 | 1994 1988 | 2000 1987 | 1974 1971 |

(a) In DF-GLS and ADF, intercept and trend; AIC for lag length; (b) Coefficient for a lagged level in the ADF equation; F-test for intercept + trend break point, or Dickey-Fuller min-t in the absence of trend or intercept break; first value in cell for innovational outlier, second for additive outlier, also for *p*-values and break dates. (c) t-Statistic 1% critical value: −5.57, 5% critical value: −5.08, 10% critical value: −4.82. (d) break date for innovational outlier and additive outlier.

## Appendix B. Cointegration Details

The literature for *I*(1) variables favours the trace test. In the presence of near unit roots, Hjalmarsson and Österholm (2010) suggest rejecting a null hypothesis of having at most *r* cointegrating equations, and by implication at least *K-r* unit eigenvalues in the system, only if both tests reject it. This leads to a conservatively low number of cointegrating relations. This is different from the suggestion

(Jusélius 2006; Kilian and Lütkepohl 2017) to be aware of the low power of unit root tests and be conservative against the hypothesis of having a unit root in case of *I*(1) variables; one would choose a low number of unit eigenvalues, *K−r*, and therefore a high number of cointegrating equations, *r*. Ever since the onset of unit-root and cointegration analysis, there has been opposition to the idea of unit roots and, correspondingly, cointegration analysis may be superfluous (Maddala and Kim 1998). Recent doubts are based more on short-term analyses of business cycles where long-term relations are seen not as additional information but as constraints (Gospodinov et al. 2013). However, they also accept the trace test as a method to determine whether there are long-term relations or a VAR in levels. This raises the question of how far low power of trace and maximum-eigenvalue tests can be used to reject all hypotheses of "at most *r < K* cointegrating equations", or, in contrast, there may be overrejection of these hypotheses and we should have low numbers *r* (Hjalmarsson and Österholm 2010). Rejecting them all leads to estimation of VARs in levels.

For the underlying VAR model, the optimal lag length is four according to the FPE, Akaike and Hannan–Quinn criteria, three according to LR; for the SIC we find two lags. The VAR model is stable with four, three or two lags. The trace test and the maximum eigenvalue test for the number of cointegrating equations show the results shown in Table A2.

**Table A2.** Johansen cointegration test: *p*-values and eigenvalues.

| Hypothesis Test | No Cointeg. Equations | At Most 1 | At Most 2 | At Most 3 | At Most 4 | At Most 5 |
|---|---|---|---|---|---|---|
| Trace | 0 | 0 | 0 | 0.0373 | 0.2122 | 0.4051 |
| Max eigenv. | 0 | 0 | 0 | 0.0882 | 0.266 | 0.4051 |
| Eigenvalue | 0.85 | 0.76 | 0.624 | 0.37 | 0.237 | 0.119 |

Series: LGDP, LTFP, LBERDST, LFBERDST, LPUBST, LFPUBST. Sample (adjusted): 1967–2017. Included observations: 51 after adjustments. Trend assumption: Linear deterministic trend (restricted). Exogenous series: DUM96 DUM13. Lags interval (in first differences): 1 to 3. MacKinnon et al. (1999) *p*-values. Trace test indicates 4 cointegrating equations at the 0.05 and 0.1 levels. Max-eigenvalue test indicates 3 (4) cointegrating equations at the 0.05 (0.1) level.

The hypotheses of no, at most 1 or at most 2 cointegrating equations are rejected by both tests. The hypothesis of three cointegrating equations is rejected by the trace test at the 5 or 10% level; the max-eigenvalue test suggests it at the 5% level but rejects it at the 10 percent level. The hypotheses of at most 4 cointegrating equations is suggested by the trace test at the 5 and 10 percent level and indicated by the max-eigenvalue at the ten percent level. Abandoning the argument of low power for our case of near-unit roots, at the ten percent level we would choose four cointegrating equations. At the five percent level we could follow the advice of Hjalmarsson and Österholm (2010) and not reject the hypothesis of three cointegrating equations, because only one test rejects it. For accepting rank 5 = *r* = *K − 1*, we would accept significance levels of 0.22 and 0.27 because of the low power of these tests. Estimating five cointegrating equations would lead to pairwise relations. Estimating a VAR in levels instead would require accepting a higher error probability of about 0.4 because of the low power of the tests.

Testing for cointegration pairwise as suggested by Kilian and Lütkepohl (2017) for I(1), we find cointegrated pairs of variables for TFP-BERDST, BERDST-PUBST, PUBST-FPUBST, FPUBST-GDP, GDP-FBERDST, all in logs, and with *p*-values between 0.15 and 0.25 in the trace (see Table 1). Surprisingly, LFPUBST and LFBERST are not cointegrated with *p*-values of 0.34 and 0.105, respectively. Together this would imply that they can be estimated as vector-error-correction model with *r* = 5 cointegrating equations if they were I(1) and with unclear recommendation for near unit roots. By implication, there would be only one unit root in the system, *K − r* = 6 − 5 = 1. However, the trace test and having near-unit roots suggest that the hypothesis of *r* = 5 should be rejected. More information for testing is needed. We can use the case *r* = 5 from the main text for comparison with models having three or four cointegrating relations below, also because pairwise relations have a nice economic interpretation.

**Appendix C. First-Difference Terms as Equilibrium Version of the Estimated VECM Model**

As we focus on the long-run growth aspects, we do not discuss results related to the differenced terms in the main text with one exception. Therefore, we put the differenced terms and statistical measures into this appendix for the sake of full information. Abbreviations: LBERDST, LFBERDST, LPUBST, LFPUBST, LGDP, LTFP are the natural logs of domestic and foreign private and public R&D stocks, gross domestic product and total factor productivity. If the long-term relations are in equilibrium in Equations (3)–(7), *CE(i)* = 0, the system is as follows.

| D(LTFP) | D(LBERDST) | D(LPUBST) | D(LFPUBST) | D(LGDP) | D(LFBERDST) |
|---|---|---|---|---|---|
| D(LTFP(−1)) | −1.473017 | −0.750921 | 0.813472 | 0.006889 | −1.942158 | −0.015236 |
| | (0.79493) | (0.49591) | (0.31552) | (0.15381) | (0.90171) | (0.19044) |
| | [−1.85302] | [−1.51423] | [2.57822] | [0.04479] | [−2.15387] | [−0.08000] |
| D(LTFP(−2)) | −1.378858 | −1.466363 | 0.334274 | −0.080010 | −1.773803 | −0.439539 |
| | (0.79501) | (0.49596) | (0.31555) | (0.15383) | (0.90181) | (0.19046) |
| | [−1.73438] | [−2.95659] | [1.05933] | [−0.52012] | [−1.96694] | [−2.30774] |
| D(LTFP(−3)) | 0.162058 | 0.021131 | 0.276971 | 0.081857 | 0.217982 | 0.045294 |
| | (0.69671) | (0.43464) | (0.27653) | (0.13481) | (0.79030) | (0.16691) |
| | [0.23261] | [0.04862] | [1.00158] | [0.60721] | [0.27582] | [0.27137] |
| D(LBERDST(−1)) | 0.440968 | 0.757832 | −0.083943 | 0.027450 | 0.588813 | 0.012469 |
| | (0.40664) | (0.25368) | (0.16140) | (0.07868) | (0.46126) | (0.09742) |
| | [1.08443] | [2.98739] | [−0.52009] | [0.34887] | [1.27653] | [0.12799] |
| D(LBERDST(−2)) | −0.061206 | 0.170539 | 0.052219 | 0.010755 | −0.236406 | 0.126510 |
| | (0.43974) | (0.27433) | (0.17454) | (0.08509) | (0.49880) | (0.10535) |
| | [−0.13919] | [0.62167] | [0.29918] | [0.12640] | [−0.47395] | [1.20088] |
| D(LBERDST(−3)) | 0.350605 | −0.046460 | 0.092931 | 0.023594 | 0.387288 | −0.104288 |
| | (0.36545) | (0.22798) | (0.14505) | (0.07071) | (0.41454) | (0.08755) |
| | [0.95938] | [−0.20379] | [0.64068] | [0.33367] | [0.93426] | [−1.19117] |
| D(LPUBST(−1)) | 0.735582 | 0.443396 | 0.642661 | −0.046334 | 0.843053 | 0.127155 |
| | (0.42287) | (0.26381) | (0.16784) | (0.08182) | (0.47968) | (0.10131) |
| | [1.73948] | [1.68076] | [3.82892] | [−0.56627] | [1.75754] | [1.25513] |
| D(LPUBST(−2)) | −1.176531 | −0.406026 | −0.226454 | 0.054214 | −1.482269 | −0.135491 |
| | (0.42863) | (0.26740) | (0.17013) | (0.08294) | (0.48621) | (0.10269) |
| | [−2.74485] | [−1.51843] | [−1.33107] | [0.65368] | [−3.04862] | [−1.31945] |
| D(LPUBST(−3)) | 0.262465 | −0.096724 | 0.036400 | −0.099835 | 0.230811 | −0.101310 |
| | (0.31682) | (0.19764) | (0.12575) | (0.06130) | (0.35937) | (0.07590) |
| | [0.82844] | [−0.48939] | [0.28946] | [−1.62859] | [0.64226] | [−1.33477] |
| D(LFPUBST(−1)) | −1.465028 | −1.255798 | −0.350081 | 0.621725 | −1.549172 | 0.427905 |
| | (1.26280) | (0.78779) | (0.50122) | (0.24434) | (1.43243) | (0.30253) |
| | [−1.16014] | [−1.59408] | [−0.69846] | [2.54449] | [−1.08150] | [1.41442] |
| D(LFPUBST(−2)) | 2.053474 | 1.031291 | 0.472678 | −0.348314 | 2.576915 | −0.466600 |
| | (1.40493) | (0.87645) | (0.55763) | (0.27184) | (1.59365) | (0.33658) |
| | [1.46162] | [1.17666] | [0.84765] | [−1.28131] | [1.61699] | [−1.38630] |
| D(LFPUBST(−3)) | −3.229958 | −0.456770 | −0.268731 | −0.017361 | −3.900740 | 0.354862 |
| | (1.12791) | (0.70364) | (0.44768) | (0.21824) | (1.27942) | (0.27021) |
| | [−2.86367] | [−0.64916] | [−0.60027] | [−0.07955] | [−3.04884] | [1.31326] |
| D(LGDP(−1)) | 1.551809 | 1.034306 | −0.600743 | 0.019399 | 1.837188 | 0.057341 |
| | (0.79890) | (0.49839) | (0.31709) | (0.15458) | (0.90622) | (0.19139) |
| | [1.94243] | [2.07530] | [−1.89452] | [0.12550] | [2.02731] | [0.29960] |
| D(LGDP(−2)) | 1.307990 | 1.294019 | −0.083448 | 0.062646 | 1.495243 | 0.317541 |
| | (0.75231) | (0.46932) | (0.29860) | (0.14557) | (0.85336) | (0.18023) |
| | [1.73864] | [2.75722] | [−0.27946] | [0.43036] | [1.75218] | [1.76186] |
| D(LGDP(−3)) | 0.058895 | −0.029050 | −0.133908 | −0.096507 | −0.016041 | −0.071775 |
| | (0.70918) | (0.44242) | (0.28148) | (0.13722) | (0.80444) | (0.16990) |
| | [0.08305] | [−0.06566] | [−0.47573] | [−0.70330] | [−0.01994] | [−0.42245] |

| | | | | | | |
|---|---|---|---|---|---|---|
| D(LFBERDST(−1)) | −1.288669 | −0.421379 | −0.277412 | −0.094861 | −1.939679 | 0.673374 |
| | (0.85077) | (0.53075) | (0.33768) | (0.16462) | (0.96506) | (0.20382) |
| | [−1.51470] | [−0.79393] | [−0.82152] | [−0.57625] | [−2.00991] | [ 3.30376] |
| D(LFBERDST(−2)) | −0.038445 | 0.476176 | 0.022386 | 0.246485 | 0.347866 | −0.054771 |
| | (1.04397) | (0.65127) | (0.41437) | (0.20200) | (1.18420) | (0.25011) |
| | [−0.03683] | [0.73115] | [0.05402] | [1.22022] | [0.29375] | [−0.21899] |
| D(LFBERDST(−3)) | −2.490940 | −0.623034 | 0.172032 | −0.069436 | −3.499444 | −0.218439 |
| | (0.89454) | (0.55805) | (0.35505) | (0.17309) | (1.01470) | (0.21431) |
| | [−2.78461] | [−1.11645] | [0.48453] | [−0.40116] | [−3.44875] | [−1.01929] |
| C | 0.093034 | −0.014553 | 0.032592 | 0.019087 | 0.160230 | 0.003539 |
| | (0.05274) | (0.03290) | (0.02093) | (0.01020) | (0.05982) | (0.01263) |
| | [1.76406] | [−0.44234] | [1.55699] | [1.87041] | [2.67839] | [0.28008] |
| DUM96 | −0.006613 | −0.016577 | 0.017396 | −0.001350 | −0.002797 | −0.004137 |
| | (0.01271) | (0.00793) | (0.00505) | (0.00246) | (0.01442) | (0.00305) |
| | [−0.52015] | [−2.09003] | [ 3.44726] | [−0.54869] | [−0.19396] | [−1.35816] |
| DUM13 | 0.044408 | 0.027929 | −0.008401 | 0.002148 | 0.056505 | 0.009046 |
| | (0.01306) | (0.00814) | (0.00518) | (0.00253) | (0.01481) | (0.00313) |
| | [3.40150] | [ 3.42921] | [−1.62131] | [0.85037] | [3.81558] | [2.89217] |
| R-squared | 0.859830 | 0.984454 | 0.993251 | 0.967802 | 0.926455 | 0.967858 |
| Adj. R-squared | 0.719659 | 0.968908 | 0.986502 | 0.935605 | 0.852911 | 0.935716 |
| Sum sq. resids | 0.003124 | 0.001216 | 0.000492 | 0.000117 | 0.004019 | 0.000179 |
| S.E. equation | 0.011178 | 0.006973 | 0.004437 | 0.002163 | 0.012679 | 0.002678 |
| F-statistic | 6.134176 | 63.32537 | 147.1688 | 30.05810 | 12.59717 | 30.11201 |
| Log likelihood | 174.9996 | 199.0643 | 222.1255 | 258.7680 | 168.5715 | 247.8737 |
| Akaike AIC | −5.843120 | −6.786835 | −7.691196 | −9.128158 | −5.591038 | −8.700928 |
| Schwarz SC | −4.858268 | −5.801983 | −6.706344 | −8.143305 | −4.606186 | −7.716076 |
| Mean dependent | 0.010904 | 0.061460 | 0.041079 | 0.029655 | 0.030934 | 0.034193 |
| S.D. dependent | 0.021111 | 0.039546 | 0.038187 | 0.008523 | 0.033060 | 0.010562 |
| Determinant resid covariance(dof adj.) | $2.67E^{-30}$ | | | | | |
| Determinant resid covariance | $3.71E^{-32}$ | | | | | |
| Log likelihood | 1410.897 | | | | | |
| Akaike information criterion | −47.83911 | | | | | |
| Schwarz criterion | −40.60423 | | | | | |
| Number of coefficients | 191 | | | | | |

## *Appendix C.1. Appendix Alternative Models*

The Johansen test would favor four or three cointegrating equations and therefore is a traditional alternative to our use of five long-term relations. Therefore, we present them, too.

## *Appendix C.2. The Model with Four Cointegrating Equations*

The final model is stable. The long-term relations are as follows.

$$CE1 = E(u_1(−1)) = 0 = LTFP(−1) − 0.062LBERDST(−1) − 0.176LPUBST(−1) − 0.00164t + 3.00$$
$$[−3.5] \qquad [−8.53] \qquad [−1.72]$$

$$CE2 = E(u_2(−1)) = 0 = LBERDST(−1) −1.09\ LPUBST(−1) − 2.01LFPUBST(−1) + 0.027t + 22.2$$
$$[−27.6] \qquad [−4.65] \qquad [2.12]$$

$$CE3 = E(u_3(−1)) = 0 = LPUBST(−1) − 0.446\ LFPUBST(−1) −1.5\ LGDP(−1) + 0.021t + 15.0$$
$$[−6.53] \qquad [−40.3] \qquad [ 8.54]$$

$$CE4 = E(u_4(−1)) = 0 = LFBERDST(−1) − 0.2LFPUBST(−1) − 0.398LGDP(−1) − 0.021t − 2.99$$
$$[−1.54] \qquad [−24.0] \qquad [−5.44]$$

TFP depends on both private and public R&D. Private R&D depends on domestic and foreign public R&D. Domestic public R&D depends on foreign public R&D and GDP. Foreign private R&D depends on foreign public R&D and reacts to Japanese GDP. T-values are reasonably large. For the last

cointegrating equation, testing says that there is one cointegrating equation for the three variables; however, this can be the relation between GDP and FBERDST reported above. For the other three cointegrating equations, we find that their three variables have two cointegrating equations. The case of four cointegrating equations, therefore, is weakened through this information.

*Appendix C.3. The Model with Three Cointegrating Equations*

The model is stable. We normalize a high slope coefficient in each relation to unity.

CE1 = E(u$_1$(−1)) = 0 =
    LFPUBST(−1) + 0.89LPUBST(−1) − 0.56LBERDST(−1) − 0.435LGDP(−1) − 0.009t - 8.07
                    [33.5]     [−23.8]      [−10.4]      [−6.49]

CE2 = E(u$_2$(−1)) = 0 =
    LBERDST(−1) −1.366LPUBST(−1) − 2.169LFPUBST(−1) + 1.15LTFP(−1) + 0.0276t + 27.3
                    [−31.98]     [−45.7]      [11.44]      [10.96]

CE3 = E(u$_3$(−1)) = 0 =
    LFBERDST(−1) − 0.945LFPUBST(−1) −0.72LPUBST(−1) − 0.42LBERDST(−1) −0.0149t + 2.4
                    [−17.05]     [−18.37]      [13.27]      [−6.67]

Foreign public R&D reacts positively to Japanese growth and private R&D, and negatively to its public R&D. Japanese private R&D reacts positively to domestic and foreign public R&D and to TFP weaknesses. Foreign private R&D reacts positively to domestic and foreign public R&D and domestic private R&D. All variables have high t-values. However, when testing for each of these sets of four variables whether they consist of exactly one cointegrating equation, the trace and maximum-eigenvalue tests suggest that there are two or three cointegrating equations; only for the second equation the max-eigenvalue test suggests that there is one cointegrating equation at the five-percent level, but at the six-percent level there would be two cointegrating equations.

There are four special cases discussed in the literature, which do not appear here. First, a variable may not depend on any long-term relation (weakly exogenous). Second, there may be one or more cointegrating equations, which do not feed into any variable and lead to the suggestion of a lower number of cointegrating equations than the trace test would suggest (Jusélius 2006, chapter 8). Third, the special case of long-run exclusion, defined as one or more of the six variables not appearing in any long-run relation by testing or imposing it (Kurita 2020), does not appear in any of the three models because all t-values indicate statistical significance. Fourth, we do not find a case β = (1, 0, 0), which would point to having I(0) variables (Fisher et al. 2016; Patterson 2000).

*Appendix C.4. Comparing the Models*

Pairwise cointegration would suggest that we have five cointegrating equations. The trace test would suggest four cointegrating equations at the five- and ten-percent levels. Near unit roots would suggest that the lower number of cointegration is not rejected by the maximum-eigenvalue test nor the trace test should be chosen. This would suggest choosing three cointegrating equations. Therefore, we need more information to decide. This information is collected in Table A3.

**Table A3.** Properties of VECMs.

| No. of Cointegrat. Eqs. | Three | Four | Five |
|---|---|---|---|
| Normal distr. (a) | 0.0423 | 0.2321 | 0.0592 |
| Serial correl. (b) | Lag 1 | Lags 1, 2, 3, 4 | Lags 1, 2 |
| Heteroscedast. (c) | 0.2556 | 0.2613 | 0.3003 |
| Adj. R-sq. (d) | 0.837, 0.67, 0.91, 0.96, 0.985, 0.94 | 0.85, 0.725, 0.94, 0.966, 0.986, 0.94 | 0.85, 0.72, 0.936, 0.97, 0.986, 0.936 |
| Log likelihood | 1392.489 | 1403.463 | 1410.897 |
| AIC | −48.137 | −48.057 | −47.839 |
| SIC | −41.89 | −41.315 | −40.604 |
| Unstable models: end year, start year | None, 1969 | 2014, 1969, 1970 | 2014, 1969, 1970 |
| No. of different signs in long-term relations (e) | 2016: 1 of 15; 2015: 1; 2014: 8; 1968: 6;1969: 6; 1970: 10; Sum:32 | 2016: 2; 2015: 3; 2014: 4; 1968: 0; 1969:1; 1970:2. Sum: 12. | 2016:3; 2015: 0; 2014: 0; 1968:0; 1969:0; 1970:0. Sum: 3. |
| No. of different sign in adjustment coefficients (e) | 2016: 0 of 17; 2015: 1; 2014: 5; 1968:11; 1969: 11; 1970: 10. Sum: 38 | 2016: 0 of 20; 2015: 3; 2014: 4; 1968: 0; 1969: 1; 1970: 2. Sum: 10. | 2016: 8; 2015: 0; 2014: 0; 1968: 0; 1969:1; 1970: 1. Sum: 10. |
| Cointegration test per set of variables in a cointegrating equation (f) | Each quadruple of any ce has more than one ce. | Each triple of any ce has more than one ce | Five pairs of cointegrating equations |

(a) Doornik–Hansen test. (b) Lagrange multiplier test at the five percent level. (c) No cross terms. (d) Order of equations: D(LGDP), D(LTFP), D(LFBERDST), D(LBERDST), D(LPUBST), D(LFPUBST). (e) for end years 2016, 2015, 2014 or start years 1968, 1969 1970 each compared to full estimation period 1967–2017. (f) See also Table 1.

Normality tests would favor four cointegrating equations. Serial correlation is unsatisfactory for models. Heteroscedasticity is no problem in any model. Models differ in a very limited way with regard to log likelihood, AIC, and SIC. Fukao and Kwon (2006) suggests recursive tests for parameter stability. They have been developed for models with more observations then we have. However, we can cut successively one, two or three observations at the beginning or the end of the sample. Parameter stability turns out to be relatively weak compared to what Soete et al. (2020a, 2020b) find in their papers. Dropping one, two or three observations at the end or the beginning of the sample leads to unstable models when ending in 2014, coming close to having the 2007−2013 crisis at the end of the sample. Moreover, starting the sample only in 1969 or 1970 may make models unstable, perhaps because they move the first oil crisis to the end of the sample. More importantly, there are many sign changes in the long-term relations when dropping observations compared to using the full sample. In this regard, the model with five cointegrating equations performs best. As discussed above, the models with three or four cointegrating equations consisting of quadruples or triples of variables fail to have exactly one cointegrating equation, but rather have pairwise or triple-wise subsets of cointegrating equations, disqualifying them.

We also experimented with a Bayesian VAR. Using Litterman/Minnesota or normal-Wishart priors leads to a negative effect of shocks on private R&D to TFP and GDP. Using Sims-Zha (normal Wishart) priors makes results extremely sensitive to the coefficient prior for initial observations dummies in a way that higher values of the coefficient prior lead to a stronger impact of private R&D on TFP for nearly equal initial shocks. Such strong dependence is undesirable (Pesaran 2015) and therefore our sample of about 50 seems to be too small for a Bayesian approach.

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
