# Peer review of "Japan’s Productivity and GDP Growth: The Role of Private, Public and Foreign R&D 1967–2017"

_economies, doi:10.3390/economies8040077_

Round 1
Reviewer 1 Report
I strongly appreciate comprehensive and rigorous application of the co-integration methodology (especially very careful examination of unit roots issues usually neglected in similar work), which leaves no doubt that empirical findings are flawless.
Nevertheless, at the moments study feels as an econometric exercise rather than scientific article, being overburden with empirics and technical details and detached from theory. Organization of the paper is unusual, over-partitioned into 9 chapters, sections are fragmented and storyline of the analysis is vague. I will briefly point out some issues that unnecessarily diminish very good overall quality of the paper and give a couple of suggestions on possible improvements in clarifying storyline, presentation of the results and organization of the paper.
- Introduction briefly provides motivation, research questions and content of the paper (which is even not necessary). Some standard elements of introduction section, such as discussion of problem background, relevance of the research and similar work are missing. The latest is especially important since no Literature Review section appears in the paper (it is legitimate also to incorporate LR in Introduction). I would advise authors to check some of the recent empirical studies published in the Economies to get clearer idea how Introduction section can be substantially improved. In addition, I suggest more precise explanation of the study storyline in the Introduction section, to make it clear how each section of the paper contributes to the listed research questions.
- In the Data section, only sources and data definition are discussed. Descriptive statistics and stylized facts that should provide initial empirical insight into the storyline are missing. The time scope of the sample is large – almost 50 years, and multiple structural breaks likely exist in the variable dynamics. In addition, I understood that dataset about R&D is not publicly available, so these data couldn’t be considered as a “common knowledge”. Hence, presentation of the key data characteristics would be very be beneficial to clarify empirical side of the storyline, especially for those readers who are not familiar with the historical development of the Japanese economy.
- The sections 4-8 are too lengthy and overburden with technical details. I would suggest to integrate sections 5 to 8 into one section, introduce the Appendix section, reconsider which empirical results from sections 4-8 are really crucial and of the utmost important to appear in body of paper, and move the rest of the results to appendix. For instance, I don’t see any value added to the paper from the detailed presentation of estimation results for each VECM model in Section 4.
Reviewer 2 Report
This is a very interesting topic. I think the authors should improve it before publication in Economies. My only concern is that the paper is too long. The authors should briefly describe the standard methods in Section 3 (VAR, cointegration etc). Also they should remove the Tables about Unit roots and Johansen tests from Section 4 (they just need to give the findings). The authors should give a focus to the main research hypotheses and results of the paper, i.e. after Section 4.4. For a good presentation of results see the following paper as an example: "Soete, Luc, Bart Verspagen and Thomas Ziesemer (2020a), The productivity effect of public R&D in the Netherlands. Economics of Innovation and New Technology 29(1): 31-47."
